# From Narratives to Diagnosis: A Machine Learning Framework for Classifying Sleep Disorders in Aging Populations: The *sleepCare* Platform

**DOI:** 10.3390/brainsci15070667

**Published:** 2025-06-20

**Authors:** Christos A. Frantzidis

**Affiliations:** School of Engineering and Physical Sciences, University of Lincoln, Lincoln LN6 7TS, UK; cfrantzidis@lincoln.ac.uk

**Keywords:** aging, BERT, machine learning, narrative analysis, natural language processing (NLP), sleep disorders, symptom detection, text classification

## Abstract

**Background/Objectives**: Sleep disorders are prevalent among aging populations and are often linked to cognitive decline, chronic conditions, and reduced quality of life. Traditional diagnostic methods, such as polysomnography, are resource-intensive and limited in accessibility. Meanwhile, individuals frequently describe their sleep experiences through unstructured narratives in clinical notes, online forums, and telehealth platforms. This study proposes a machine learning pipeline (***sleepCare***) that classifies sleep-related narratives into clinically meaningful categories, including stress-related, neurodegenerative, and breathing-related disorders. The proposed framework employs natural language processing (NLP) and machine learning techniques to support remote applications and real-time patient monitoring, offering a scalable solution for the early identification of sleep disturbances. **Methods**: The ***sleepCare*** consists of a three-tiered classification pipeline to analyze narrative sleep reports. First, a baseline model used a Multinomial Naïve Bayes classifier with n-gram features from a Bag-of-Words representation. Next, a Support Vector Machine (SVM) was trained on GloVe-based word embeddings to capture semantic context. Finally, a transformer-based model (BERT) was fine-tuned to extract contextual embeddings, using the [CLS] token as input for SVM classification. Each model was evaluated using stratified train-test splits and 10-fold cross-validation. Hyperparameter tuning via GridSearchCV optimized performance. The dataset contained 475 labeled sleep narratives, classified into five etiological categories relevant for clinical interpretation. **Results**: The transformer-based model utilizing BERT embeddings and an optimized Support Vector Machine classifier achieved an overall accuracy of **81%** on the test set. Class-wise F1-scores ranged from **0.72 to 0.91**, with the highest performance observed in classifying **normal or improved sleep** (F1 = 0.91). The **macro average F1-score** was **0.78**, indicating balanced performance across all categories. GridSearchCV identified the optimal SVM parameters (C = 4, kernel = ‘rbf’, gamma = 0.01, degree = 2, class_weight = ‘balanced’). The confusion matrix revealed robust classification with limited misclassifications, particularly between overlapping symptom categories such as stress-related and neurodegenerative sleep disturbances. **Conclusions**: Unlike generic large language model applications, our approach emphasizes the **personalized identification of sleep symptomatology** through targeted classification of the narrative input. By integrating structured learning with contextual embeddings, the framework offers a **clinically meaningful**, scalable solution for early detection and differentiation of sleep disorders in diverse, real-world, and remote settings.

## 1. Introduction

Sleep is a fundamental biological mechanism with a pivotal role in preserving cognitive, physical wellbeing, and proper emotion regulation [1]. However, aging populations may experience sleep disorders related to insomnia [2] or sleep fragmentation [3] due to stress-related factors [4], neurodegeneration [5], or other chronic conditions [6] and poor sleep quality [7]. There is now concrete evidence from epidemiological studies that more than half of older adults report sleep disturbances [8]. Apart from being detrimental to quality of life, the sleep barriers may also be indicative of physical and mental illnesses progression such as neurodegeneration [5] and mental disorders [9].

As we age, there are sleep architecture alterations such as reduced slow wave (N3) and Rapid Eye Movement (REM) duration, which is often accompanied by more frequent awakenings and increased sleep latency [10]. Those disturbances are commonly associated with comorbidities such as chronic conditions (cardiovascular disease, diabetes, chronic pain) [6], psychiatric disorders (depression, anxiety) [11], and lifestyle factors such as sedentary behavior and lack of activity [12]. Importantly, emerging evidence suggests that sleep disturbances may precede and potentially contribute to the development of neurodegenerative disorders by accelerating the accumulation of pathological proteins (e.g., amyloid-beta and tau in AD) and impairing clearance mechanisms such as glymphatic flow [13]. There is concrete evidence that irregular work schedules and shiftwork induce sleep disorders in shift-working nurses. A previous study employed the Bergen Shft Work Sleep Questionnaire (BSWSQ) to survey 1586 Norwegian nurses and found that night shifts and rotating schedules were most associated with insomnia and sleepiness. Moreover, three-shift rotations induced a higher insomnia rate than permanent night shifts [14].

Given the growing aging population and the rising incidence of neurodegenerative diseases, understanding the bidirectional relationship between sleep disorders and neurodegeneration is critical. Interventions aimed at improving sleep quality through behavioral, environmental, or pharmacological means may offer promising avenues to delay or mitigate the onset and progression of cognitive impairment in older adults [15].

The concept of Natural Language Processing (NLP) enables the analysis of unstructured text from clinical notes, Electronic Health Records (EHRs), and patient queries, which results in the automatic extraction of sleep-related information. Sleep text classification is a technique that receives as input narrative data from users and can identify a wide range of sleep-related symptoms such as insomnias, sleep apneas, parasomnias, and hypersomnias.

Early studies in that field aimed at the identification of publication trends in sleep disorder research using text mining methods [16]. It performed text mining on 4515 PubMed articles through cluster analysis and logistic regression. Although it was a pioneering NLP study in the fields of sleep-related textual data, it suffered from several limitations, especially when dealing with ambiguous terminology. NLP techniques were also used to extract sleep-related symptomatology from EHRs, and their results are summarized in a systematic review that compared 27 NLP studies on EHR free-text narratives [17]. The authors concluded that most of the studies focus on developmental aspects and not on the symptomatology itself. NLP techniques such as topic modeling and sentiment analysis can also be combined with traditional outcome measures and provide hybrid approaches in evaluating the efficacy of sleep-related interventions [18]. Recently, the use of Large Language Models (LLMs) has induced revolutionary changes in the NLP field. Many patients refer to publicly available LLM models for medical advice. A recent blinded study compared ChatGPT and sleep specialists’ responses, rated by experts and laypeople [19]. The main finding of this study was that ChatGPT was rated higher for emotional supportiveness and clarity by laypeople, whereas it yielded slightly lower accuracy rated by experts. The main finding of the study was that ChatGPT responses may contain inaccuracies and a lack of real-world clinical context.

Although there is concrete evidence of the association of sleep disorders with cognitive decline and physical and mental wellbeing, the identification of sleep disorders is mainly performed in clinical environments through polysomnography [20] or in real-world settings with wearables and or mobile applications [21]. Most methodological approaches remain limited by time-intensive diagnostic protocols, subjective self-reports, and the fragmented integration of sleep symptomatology across health systems [22]. Traditional methods have not, until now, integrated the enormous wealth of the narratives that patients use to describe their sleep issues. Although these are highly subjective, these narratives may implicitly reveal the underlying causes of sleep disturbances, such as stress, neurodegeneration, poor sleep hygiene, or physiological conditions like sleep apnea [23].

The rationale of the present study is to contribute to the research agenda by bridging a significant gap between qualitative symptom expression and structured clinical interpretation. Currently, existing NLP approaches in healthcare focus on diagnosis or named-entity recognition without accounting for the latent semantic cues that suggest cause-specific sleep disturbances. Our aim is to integrate a publicly available, custom-generated dataset with word embeddings and transformer architectures to offer high-resolution insight into symptom patterns that would otherwise remain unclassified or misinterpreted.

The proposed pipeline represents a scalable, non-invasive, and efficient alternative enabling the early detection and differentiation of sleep problems across diverse and/or remote populations. Firstly, baseline models founded on naïve techniques were investigated, and then the classification accuracy was further improved by employing word embeddings and transformer-based models. The latter concepts improved the robustness of our approach, making it capable of processing unstructured textual input derived from patient self-reports, electronic health records, and chatbot conversations and classifying them into five clinically meaningful categories. These categories not only differentiated between normal and disordered sleep but also identified probable etiologies: (1) stress-related factors, (2) neurodegenerative processes, (3) breathing-related disorders, and (4) poor sleep habits.

## 2. Materials and Methods

### 2.1. Dataset

The author constructed a dataset consisting of narrative, sleep-related text samples in sentence form. The text samples were derived from electronic sources describing the causes and factors of the main sleep disorders, such as the National Sleep Foundation (NSF) [24], the American Academy of Sleep Medicine (AASM) [25], Centers for Disease Control and Prevention (CDC)—Sleep and Sleep Disorders [26], National Institutes of Health (NIH)—Sleep Disorders Information [27], and World Sleep Society [28]. Each sentence is labeled according to the underlying reason for sleep quality or disturbance. The dataset contains a total of **623** text entries, where each entry is a free-form narrative describing an individual’s sleep experience. Each entry includes two fields, as follows:•The first (text) is a natural language description of a sleep-related experience;•The second (label) is an integer from 0 to 4, representing the primary cause or nature of the sleep condition.

The label encoding is as follows:•0—Sleep disorders related to stress-related factors (e.g., interpersonal conflict, job insecurity, grief, emotional distress);•1—Sleep disturbances due to neurodegeneration or chronic conditions (e.g., dementia, diabetes, hypertension);•2—Breathing-related sleep disorders, especially sleep apnea and its effects (e.g., snoring, gasping, CPAP use);•3—Sleep disruption due to poor sleep habits or environmental/lifestyle factors (e.g., noise, caffeine, light exposure, irregular schedules);•4—Normal or improved sleep experiences or narratives of recovered/restful sleep.

The dataset consists of (1) 43 instances (6.9%) of sleep disorders with stress/emotional issues, (2) 156 instances (25%) of sleep disorders due to medical conditions, (3) 66 instances (10.6%) of breathing-related sleep disorders, (4) 98 instances (15.7%) of poor sleep habits attributed to environmental/behavioral factors in accordance with a study investigating shiftwork in Norwegian nurses [14], and (5) 260 instances (41.7%) of good sleep quality. The dataset is moderately imbalanced, with **Category 4** (positive sleep experiences) comprising over **40%** of the sample. The first two categories (stress/emotional and breathing disorders) are the smallest, together accounting for only about **18%** of the data. This imbalance should be considered when training machine learning models, especially when using accuracy as a metric, and might require techniques like **class weighting**, **resampling**, or **stratified cross-validation**. The distribution of the text samples across the different categories is visualized in Figure 1, as follows:

The dataset aims to integrate scientific evidence associated with good sleep practices and sleep-associated disorders as defined by world-leading organizations [24,25,26,27,28]. Then, it transforms those guidelines into patient-related narratives. It is an ongoing procedure that is currently not balanced across all the classification categories. Therefore, there is a need for further enhancing the sleep disorders and especially those associated with mental health factors. Further dataset versions should also integrate the existing surrogate gate with real-world instances from lived experience. It is publicly available at the following link [29] and can be applied to several research and development areas, including

•Evaluation of machine learning algorithms for the automated classification of sleep disturbances from narrative text;•Identification of thematic patterns in personal sleep narratives;•Training of models capable of distinguishing between multiple etiologies of sleep disturbance;•Development of personalized sleep health interventions and digital mental health tools that rely on natural language input from users.

### 2.2. Pre-Processing

The author employed several resources from the Natural Language Processing Toolkit (NLTK) library to pre-process the textual data as follows:•The Punkt Tokenizer Models were used for sentence splitting and word tokenization. We used it to split a paragraph into sentences or words;•The ‘stopwords’ was used to download a list of common stopwords (‘the’, ‘is’, ‘and’, etc.). We use this to filter out common words that usually carry less meaning in text analysis;•The WordNet lexical database was used for lemmatization, which reduces words to their base form;•The ‘averaged_perceptron_tagger’ uses the part-of-speech (POS) tagger. This tag words in a sentence with their grammatical role (noun, verb, adjective, etc.).

The pre-processing code accomplished the following tasks: (1) lowercasing, (2) removing punctuation, (3) tokenization, (4) stopword removal, (5) POS tagging, and (6) lemmatization.

### 2.3. Baseline Model

Firstly, a baseline model was established by implementing a supervised machine learning pipeline using a Multinomial Naïve Bayes classifier. The dataset was first preprocessed using the Natural Language Toolkit (NLTK). The pre-processing involved the removal of stopwords and punctuation points. Then, the text was tokenized to retain only semantically relevant tokens. The cleaned dataset was then split into training (80%) and testing (20%) sets using stratified random sampling to ensure balanced representation across categories.

The textual feature extraction was performed through the Bag-of-Words model via CountVectorizer. The author experimented with varying n-gram ranges from unigrams to 6-g. For each n-gram configuration, a separate Multinomial Naive Bayes classifier was trained on the resulting feature vectors and evaluated on the test set. Model performance was assessed using classification accuracy. The accuracy results were plotted to visualize the relationship between n-gram size and predictive performance. This approach provided a scalable and interpretable framework for detecting sleep disorder patterns in narrative data. It formulated the baseline model and the groundwork for more advanced methods, such as contextual word embeddings and transformer-based models.

### 2.4. The Bidirectional Encoder Representations from Transformers (BERTs) Model

It is a deep learning model introduced by Google [30]. BERT employs the Transformer architecture to generate context-aware word embeddings. Unlike traditional models, it considers both left and right contexts simultaneously at every layer. This has revolutionized the NLP field, since it can capture deeper context and meaning from textual data. The ‘bert-base-uncased’ version, which is a 12-layer Transformer model trained on lowercased English text from BookCorpus and English Wikipedia, was used. The tokenizer pre-processes raw input text into subword tokens and converts them into integer IDs and attention masks. The model itself returns hidden states that serve as rich contextual embeddings.

The model switched to inference mode by calling the ‘.eval()’ method. This action disables dropout layers to ensure deterministic behavior. This is frequently used during feature extraction in classification tasks. The ‘outputs.last_hidden_state’ embeddings were used as input to machine learning classifiers, making BERT a powerful backbone for fine-tuned NLP pipelines.

To extract high-quality, contextualized sentence representations using BERT, the [CLS] token embedding was utilized from the model’s final hidden layer. The get_bert_embedding(text) function tokenizes input text using the bert-base-uncased tokenizer, applies truncation and padding to handle variable-length sequences, and converts the data into PyTorch (version 2.6.0) tensors. The pre-trained BERT model is then executed in inference mode using torch.no_grad() to efficiently compute hidden states without gradient tracking. From the resulting tensor outputs.last_hidden_state, which has shape [batch_size, sequence_length, hidden_size], the embedding of the [CLS] token located at index 0 was extracted. This 768-dimensional vector serves as a compact representation of the entire input sequence and is commonly used in downstream tasks such as sentence classification, clustering, or semantic similarity analysis. The function returns this vector as a NumPy array, making it directly usable in traditional machine learning pipelines or as input to further neural layers.

### 2.5. Word Embeddings

A more advanced model was next implemented through a semantically rich classifier capable of distinguishing between the various types of sleep-related narratives. To achieve this, a word embedding-based approach using GloVe vectors combined with Support Vector Machine (SVM) classification was employed. The dataset was first preprocessed in the same way as the baseline model. The same tokenization approach as before was applied to convert each entry into a sequence of lowercased tokens.

Then, the pre-trained GloVe word embeddings were utilized with 50 dimensions to represent each narrative into a numerical (vector) format. For each sentence (document), document-level embedding was computed by averaging the word vectors for each token present in the vocabulary. Entries containing no valid tokens (i.e., words not found in the embedding index) were excluded to ensure meaningful input representations.

The classification of the extracted features (word embeddings) was performed through a multi-class SVM classifier. Aiming to optimize the classification accuracy, the author conducted a comprehensive hyperparameter search using the GridSearchCV function. Various kernels (linear, rbf, poly, sigmoid), regularization values (C), kernel coefficients (gamma), and polynomial degrees were explored. A 10-fold cross-validation strategy was used during this process to identify the configuration that achieved the highest average classification accuracy across folds.

After identifying the optimal parameter set, the SVM model was retrained using the best combination on the full training data and evaluated its performance on the sequestered **test set**. This evaluation provided an unbiased estimate of generalization performance, reported via standard metrics such as accuracy, precision, recall, and the confusion matrix. This embedding-based methodology enables the efficient classification of complex sleep narratives and serves as a strong baseline for comparison with more computationally intensive deep learning models.

### 2.6. Machine Learning Algorithms Used with BERT

#### 2.6.1. Support Vector Machines (SVM) Design Methodology

A Support Vector Machine (SVM) classifier was retrained using the extracted BERT embeddings. To optimize the classifier’s performance, a comprehensive hyperparameter search was conducted using GridSearchCV from the scikit-learn library. The parameter grid included the following:•C: Regularization parameter—values of [0.1, 0.4, 1, 2, 3, **4**];•gamma: Kernel coefficient—values of [0.001, 0.01, 0.1, 0.4, 1, **2**, 3, 4, ’scale’, ’auto’];•kernel: Kernel type—values of [‘linear’, ‘**rbf**’, ‘poly’, ‘sigmoid’];•degree of the polynomial kernel—values of [2, 3, 4, 5, 6, 7, 8];•‘class_weight’: [None, ‘**balanced**’].

Different values for the regularization parameter/C, which performs a trade-off between margin maximization and minimization of the classification error, were explored. The kernel coefficient is used for the ‘rbf’, ‘poly’, and ‘sigmoid’ kernels. It determines the influence of each unique training instance. Smaller gamma values produce smoother decision boundaries, whereas higher gamma values result in more complex decision boundaries and maybe overfitting. The next parameter to fine-tune is the type of kernel function. The author investigated four different kernel types: (1) linear, (2) radial basis function, (3) polynomial, and (4) sigmoid. These specify the type (linear vs. non-linear) of kernel function used to project data into a higher-dimensional space. Higher degrees mean more complex decision boundaries. Another important SVM parameter is the class_weight. When fine-tuning this parameter, we can adjust weights for different classes. This is particularly useful in the case of class imbalance. The None value treats all classes equally, whereas the ‘balanced’ value adjusts the weights automatically in order to be inversely proportional to class frequencies in the data. The optimal parameters are indicated in bold.

The grid search was performed using 10-fold cross-validation on the training set. This process systematically evaluated all parameter combinations to identify the model configuration that achieved the highest average classification accuracy. The resulting best-performing model was retrained on the full training set before final evaluation.

The trained model was evaluated on a hold-out test set, which comprised 20% of the original data. Performance was assessed using standard classification metrics, including accuracy, precision, recall, and F1-score. Additionally, a confusion matrix was plotted to visualize the classifier’s performance across different classes. All results were computed using scikit-learn’s evaluation utilities.

#### 2.6.2. Random Forest (RF) Classifier Design Methodology

To develop a robust classification model, a Random Forest classifier with hyperparameter tuning via GridSearchCV was employed. The feature set consisted of BERT-derived embeddings extracted from preprocessed text, and the corresponding labels were retained from the original dataset. A parameter grid was defined to explore a range of configurations, including the number of trees in the forest (n_estimators ranging from 5 to 50 with a step of 5), the maximum depth of each tree (max_depth from 5 to 30 with a step of 5 and None for unconstrained growth), and the minimum number of samples required to split a node (min_samples_split: [2, 5, 7, 10]), and to define a leaf (min_samples_leaf: [1, 2, 4]). The optimal parameter set is the following: ‘bootstrap’: False, ‘max_depth’: None, ‘min_samples_leaf’: 2, ‘min_samples_split’: 2, ‘n_estimators’: 30. Both bootstrap sampling strategies (True, False) were also evaluated. A 10-fold cross-validation was applied within the grid search to ensure robust model selection based on classification accuracy. The grid search was executed in parallel using all CPU cores (n_jobs = −1), and the best-performing model was selected and evaluated on a held-out test set. The final model’s predictive performance was assessed using standard classification metrics and visualized through a confusion matrix to understand its performance across all classes.

#### 2.6.3. XGBoost Classifier Design Methodology

Finally, an XGBoost classifier trained on BERT-generated sentence embeddings was developed. The data were partitioned into training and test sets using an 80/20 split and transformed into XGBoost’s optimized DMatrix format. We applied a structured three-stage grid search approach to identify the best hyperparameter configuration. In **Stage 1**, tree structure parameters (max_depth and min_child_weight) were tuned using 5-fold cross-validation with early stopping based on multi-class log loss. The optimal values were (1) max_depth: 3 and (2) min_child_weight: 5. In **Stage 2**, we further refined the model by adjusting regularization (reg_alpha, reg_lambda) and sampling parameters (subsample, colsample_bytree). The optimal values obtained in this stage were (1) subsample: 0.7, (2) colsample_bytree: 0.7, (3) reg_alpha: 0.5, and (4) reg_lambda: 1.5. Finally, in **Stage 3**, we explored various learning rates (learning_rate) to balance convergence speed and generalization. At each stage, the configuration yielding the lowest average log loss on validation folds was selected. The final model was trained on the full training set using the best parameters and optimal number of boosting rounds. Performance was evaluated on the test set using a classification report, confusion matrix, and learning curve visualization, providing a comprehensive view of the model’s accuracy and calibration across classes. The final set of the fine-tuned parameters is the following: {‘objective’: ‘multi:softprob’, ‘num_class’: 5, ‘eval_metric’: ‘mlogloss’, ‘seed’: 42, ‘max_depth’: 3, ‘min_child_weight’: 5, ‘subsample’: 0.7, ‘colsample_bytree’: 0.7, ‘reg_alpha’: 0.5, ‘reg_lambda’: 1.5, ‘learning_rate’: 0.05}}.

## 3. Results

### 3.1. Baseline Model Evaluation

The classification performance of a Multinomial Naive Bayes model was evaluated across different n-gram configurations. Text data were preprocessed by removing English stopwords and punctuation and then split into training and testing sets with an 80/20 ratio. CountVectorizer was used to extract n-gram features, varying the upper limit of the n-gram range from one to nine. For each n-gram configuration, the model was trained on the training set and evaluated on the test set using accuracy as the performance metric.

The results are summarized in Figure 2 and Table 1. The highest test accuracy (acc = 0.84) was observed when using unigrams (*n* = 1). As the value of n increased from 1 to 4, the model’s accuracy consistently decreased from 0.840 to 0.816 for *n* = 4 (Table 1). At that point, the most pronounced drop in accuracy occurred between *n* = 1 and *n* = 2. For n-values between 4 and 6, the accuracy reached a plateau of 0.816 and further decreased to 0.808 for n-values greater than or equal to 7. Across all the tested configurations, unigrams yielded the best classification results, while higher-order n-grams resulted in reduced performance. However, it should be highlighted that the accuracy drop due to the increase in the n-value was not a dramatic one. The baseline model showed a steady performance, which was a bit higher than 80%, and was enhanced for the smallest n-values. This finding is further discussed in the remainder of the article and was associated with the small size of the individual narratives.

Aiming to investigate the classifier’s performance in more detail, precision, recall, f1-score, and support were computed for each class regarding the unigram model. The evaluation metrics are visualized in Table 2 and the classification results in Figure 3A. For class 0 (stress-related factors), the model achieved a precision of 1.00, a recall of 0.45, and an F1-score of 0.62, based on 11 test samples. For class 1 (medical conditions), the precision was 0.97, the recall was 0.80, and the F1-score was 0.88, across 35 samples. Class 2 (breathing-related sleep disorders) achieved a precision of 0.89, a recall of 0.94, and an F1-score of 0.91 over 17 samples. For class 3 (poor sleep habits), the precision was 0.64, the recall was 0.78, and the F1-score was 0.71, based on 23 samples. For class 4 (normal sleep), the model obtained a precision of 0.84, a recall of 0.97, and an F1-score of 0.90 across 39 samples. The overall accuracy of the unigram model on the test set was 0.84, computed over a total of 125 samples. The macro-averaged precision, recall, and F1-score were 0.87, 0.79, and 0.80, respectively. The weighted averages for precision, recall, and F1-score were 0.86, 0.84, and 0.84, reflecting the class distribution of the test data.

### 3.2. Word Embedding Model Evaluation

Following the n-gram experiments, word embeddings generated using pre-trained GloVe vectors were utilized as features for text classification. To evaluate the performance of the model leveraging GloVe word embeddings, the author conducted an extensive hyperparameter optimization using GridSearchCV with 10-fold cross-validation. A total of 115,200 model fits were executed across 11,520 unique hyperparameter combinations. The SVM classifier achieved its optimal performance with the following configuration: C = 3, kernel = ‘poly’, degree = 3, gamma = ‘scale’, and class_weight = None. These parameters correspond to a polynomial kernel of degree 3 without class reweighting and with the default scaling for the kernel coefficient. The selected model was subsequently used for further evaluation on the test dataset. The overall accuracy was 0.70.

For class 0 (stress-related factors), the model achieved a precision of 0.88, a recall of 0.64, and an F1-score of 0.74, based on 11 test samples. For class 1 (medical conditions), the precision was 0.77, the recall was 0.66, and the F1-score was 0.71, with 35 samples. Class 2 (breathing-related sleep disorders) yielded a precision of 0.81, a recall of 0.76, and an F1-score of 0.79 over 17 samples. For class 3 (poor sleep habits), the model achieved a precision of 0.42, a recall of 0.35, and an F1-score of 0.38, based on 23 samples. For class 4 (normal sleep), the model showed a precision of 0.69, a recall of 0.92, and an F1-score of 0.79, across 39 samples. The overall accuracy of the model on the test set was 0.70, calculated over a total of 125 samples. The macro-averaged precision, recall, and F1-score were 0.71, 0.67, and 0.68, respectively. The weighted averages for precision, recall, and F1-score were 0.70, 0.70, and 0.69, respectively, reflecting the distribution of samples across classes (Table 3).

**Table 3 brainsci-15-00667-t003:** Classification accuracy of the word embedding model.

Class	Precision	Recall	F1-Score	Support
0	0.88	0.64	0.74	11
1	0.77	0.66	0.71	35
2	0.81	0.76	0.79	17
3	0.42	0.35	0.38	23
4	0.69	0.92	0.79	39

### 3.3. BERT Transformers Model Evaluation

#### 3.3.1. Support Vector Machines

The best set of hyperparameters identified through the grid search consisted of C = 4, class_weight = ‘balanced’, kernel = ‘rbf’, gamma = 0.01, and degree = 2. In this configuration, a radial basis function (RBF) kernel was used with a manually specified gamma value and automatic balancing of class weights. This optimized model was selected for further evaluation on the test dataset.

The performance of the optimized Support Vector Machine (SVM) model was evaluated on the test set using standard classification metrics: precision, recall, F1-score, and support for each class. The results are summarized in Table 4 and Figure 4A. For class 0 (stress-related factors), the model achieved a precision of 0.73, a recall of 0.73, and an F1-score of 0.73, based on 11 test samples. For class 1 (medical conditions), the precision was 0.87, the recall was 0.74, and the F1-score was 0.80, with 35 samples. Class 2 (breathing-related sleep disorders) yielded a precision of 0.70, a recall of 0.82, and an F1-score of 0.76 over 17 samples. For class 3 (poor sleep habits), the model achieved a precision of 0.71, a recall of 0.74, and an F1-score of 0.72, based on 23 samples. For class 4 (normal sleep), the model showed a precision of 0.90, a recall of 0.92, and an F1-score of 0.91, across 39 samples. The overall accuracy of the model on the test set was 0.81, calculated over a total of 125 samples. The macro-averaged precision, recall, and F1-score were 0.78, 0.79, and 0.78, respectively. The weighted averages for precision, recall, and F1-score were all 0.81, reflecting the distribution of samples across classes. The confusion matrix is displayed in Figure 4A.

#### 3.3.2. Random Forests

A Random Forest classifier was trained using the precomputed BERT embeddings, and hyperparameter optimization was conducted via GridSearchCV. The parameter grid included variations in the number of estimators (n_estimators), maximum tree depth (max_depth), minimum number of samples required to split an internal node (min_samples_split), minimum number of samples required at a leaf node (min_samples_leaf), and bootstrap sampling (bootstrap). Totally, 1680 different hyperparameter combinations were evaluated using 10-fold cross-validation, resulting in 16,800 individual model fits. The best set of hyperparameters identified consisted of n_estimators = 30, max_depth = None, min_samples_split = 2, min_samples_leaf = 2, and bootstrap = False. This configuration, featuring fully grown trees without bootstrapping and a relatively small minimum leaf size, was selected for final evaluation.

The Random Forest classifier was evaluated on the test set using standard classification metrics: precision, recall, F1-score, and support for each class. The results are presented in Table 5. For class 0 (stress-related factors), the model achieved a precision of 0.75, a recall of 0.55, and an F1-score of 0.63, based on 11 test samples. For class 1 (medical conditions), the precision was 0.75, the recall was 0.69, and the F1-score was 0.72, across 35 samples. Class 2 (breathing-related sleep disorders) achieved a precision of 0.61, a recall of 0.65, and an F1-score of 0.63 over 17 samples. For class 3 (poor sleep habits), the model obtained a precision of 0.50, a recall of 0.39, and an F1-score of 0.44, based on 23 samples. For class 4 (normal sleep), the precision was 0.69, the recall was 0.87, and the F1-score was 0.77, across 39 samples. The overall accuracy of the Random Forest model on the test set was 0.67, evaluated over a total of 125 samples. The macro-averaged precision, recall, and F1-score were 0.66, 0.63, and 0.64, respectively. The weighted averages for precision, recall, and F1-score were 0.67, 0.67, and 0.66, respectively, taking into account the class distribution. The confusion matrix is displayed in Figure 4B.

#### 3.3.3. XGBoost

An XGBoost classifier was trained using the precomputed BERT embeddings, following a three-stage hyperparameter tuning process. In the first stage, parameters related to the tree structure were optimized. The best configuration included a maximum tree depth of 3 and a minimum child weight of 5, indicating smaller, more regularized trees. In the second stage, subsampling and regularization parameters were tuned. The optimal model utilized 70% of the available training data per boosting round (subsample of 0.7) and randomly sampled 70% of the available features per tree (colsample_bytree of 0.7). Additionally, an L1 regularization strength (reg_alpha) of 0.5 and an L2 regularization strength (reg_lambda) of 1.5 were applied to penalize overly complex models. In the third stage, the learning rate was adjusted, and the best performance was achieved with a learning rate of 0.05. The final model was trained for 464 boosting rounds, as determined by early stopping during cross-validation. The fully optimized model was evaluated on the test set using precision, recall, F1-score, and support for each class. The results are summarized in Table 6.

For class 0 (stress-related factors), the model achieved a precision of 0.60, a recall of 0.55, and an F1-score of 0.57, based on 11 test samples. For class 1 (medical conditions), the precision was 0.82, the recall was 0.77, and the F1-score was 0.79, across 35 samples. Class 2 (breathing-related sleep disorders) obtained a precision of 0.78, a recall of 0.82, and an F1-score of 0.80, across 17 samples. For class 3 (poor sleep habits), the precision was 0.64, the recall was 0.61, and the F1-score was 0.62, based on 23 samples. For class 4 (normal sleep), the model achieved a precision of 0.81, a recall of 0.87, and an F1-score of 0.84, across 39 samples. The overall accuracy of the XGBoost model on the test set was 0.76, computed over 125 samples. The macro-averaged precision, recall, and F1-score were 0.73, 0.72, and 0.73, respectively, while the weighted averages for these metrics were each 0.76. The confusion matrix is displayed in Figure 4C.

### 3.4. Comparative Analysis of Models’ Performance

A comparative analysis of the models’ performance is summarized in Table 7. Among the five models evaluated, the Unigram Naive Bayes classifier achieved the highest test accuracy (0.84) and macro-averaged F1-score (0.80). The BERT embeddings combined with a Support Vector Machine classifier also performed strongly, reaching an accuracy of 0.81 and a macro F1-score of 0.78. The XGBoost model using BERT embeddings attained an accuracy of 0.76 and a macro F1-score of 0.73, outperforming the Random Forest classifier trained on BERT embeddings, which achieved an accuracy of 0.67 and a macro F1-score of 0.64. In contrast, the GloVe-based SVM classifier achieved an accuracy of 0.70 and a macro F1-score of 0.68, performing better than Random Forest but below both the Unigram and BERT-SVM models. The code was executed on a Dell computer equipped with an Intel i7 processor and a GPU. The execution time for the simpler model, the Naive Bayes classifier, was 0.853 s, while the BERT-based methodologies required approximately 15 min.

## 4. Discussion

The current study introduces ***sleepCare***, an automated Natural Language Processing (NLP) pipeline designed for the classification of sleep-related textual data. The ***sleepCare*** approach systematically integrates NLP methodologies—including n-gram models and pre-trained word embeddings such as GloVe and BERT—with robust machine learning algorithms such as Support Vector Machines, Random Forests, and XGBoost classifiers. Drawing from best practices in modern text classification pipelines, the system incorporates essential phases such as advanced text preprocessing, feature extraction, hyperparameter tuning, and a structured model evaluation process. By transforming unstructured and highly variable text into structured, high-quality representations, ***sleepCare*** establishes a reliable framework for automating complex text categorization tasks [31].

The ***sleepCare*** pipeline currently employs traditional machine learning algorithms such as (1) Support Vector Machines, (2) Random Forests, and (3) XGBoost. The selection of the machine learning models was performed based on the following criteria: (1) to identify the ones most frequently used in NLP pipelines and (2) to employ models that allow easier hyperparameter tuning to optimize the model’s performance. As the dataset size increases, more elaborate neural (recurrent) network architectures will be investigated.

The ***sleepCare*** is regarded as a digital health application that enables the automated analysis of sleep-related textual reports such as sleep diaries, questionnaires, and patient descriptions [32]. The proposed pipeline may have important applications for remote sleep medicine solutions by providing early alerts of the most common sleep disorders [33] in underserved or rural areas in which access to specialized sleep centers and clinicians is limited [34]. So, ***sleepCare*** may enhance the arsenal of digital health tools toward preliminary screening, flag high-risk individuals for further clinical evaluation, and ultimately facilitate earlier intervention and improved management of sleep health, contributing to reduced morbidity and enhanced quality of life in populations with traditionally limited access to sleep medicine services [35].

The efficacy of the ***sleepCare*** should be further validated through a pilot phase, which would contain real-world data. The study design should integrate elderly participants from rural and urban settings, who would be randomly assigned either to a group interacting with the ***sleepCare*** system or with a gold standard, with clinical settings, such as access to sleep clinics equipped with polysomnographic sensors and sleep experts. The potential outcome measures should be the accuracy of both approaches, the ease of use, and the users’ satisfaction and adherence. Apart from those dependent variables, the classification accuracy of the ***sleepCare*** system should be further validated by means of sensorial and question-related data. Therefore, the future study design should also investigate whether the system’s approach would be statistically significantly correlated with the outcome of the Pittsburgh Sleep Quality Index (PSQI) [36] or the SmartHypnos mobile application [21].

In addition to developing the ***sleepCare*** classification pipeline, the present study contributed by constructing and publicly releasing a novel dataset of sleep-related narratives [29]. The dataset was curated following the guidelines and terminologies proposed by leading sleep organizations, including the National Sleep Foundation (NSF) [24], the American Academy of Sleep Medicine (AASM) [25], the Centers for Disease Control and Prevention (CDC) [26], the National Institutes of Health (NIH) [27], and the World Sleep Society [28]. Each narrative was carefully labeled to reflect specific symptomatology across major etiological categories of sleep disturbances, such as stress-related disorders, neurodegenerative-related sleep problems, breathing-related disorders, poor sleep hygiene, and descriptions of normal or improved sleep. By explicitly identifying probable underlying causes within naturalistic sleep narratives, the dataset addresses a critical gap in existing resources, which often overlook the nuanced expression of symptom patterns. Moreover, by following official clinical guidelines and explicitly labeling symptomatology, the construction of the ***sleepCare*** dataset advances broader explainability standards emphasized in the AI research community [37]. It ensures that the features and categories used for model training are interpretable, clinically meaningful, and transparent, addressing key concerns about trustworthiness and reproducibility in health-related natural language processing. The open availability of this dataset fosters scientific transparency, enables reproducibility, and encourages collaboration across multidisciplinary teams aiming to advance sleep disorder screening, digital sleep health interventions, and personalized sleep medicine research. It lays the groundwork for future innovations in NLP-driven sleep diagnostics, particularly in remote, aging, or underserved populations where traditional assessments are less accessible.

A comparative analysis was conducted across five different modeling approaches to assess the effectiveness of the ***sleepCare*** pipeline in classifying sleep-related narratives. Among all models, the Unigram-based Naive Bayes classifier achieved the highest overall accuracy (0.84) and macro-averaged F1-score (0.80), highlighting the strong predictive power of simple word frequency features for structured narrative datasets. However, further inspection of the class-wise precision and recall metrics revealed that its performance was disproportionately skewed toward majority classes, with poor recall for minority categories. In contrast, the BERT-based Support Vector Machine model achieved an accuracy of 0.81 and a macro-averaged F1-score of 0.78, while demonstrating more balanced performance across all classes. Notably, the SVM model incorporated a class weighting strategy during hyperparameter tuning (**class_weight = ‘balanced’**), which improved its sensitivity to less-represented sleep disorder categories and helped mitigate the effects of data imbalance. The BERT-based XGBoost model also showed competitive performance, achieving an accuracy of 0.76 and a macro F1-score of 0.73, benefiting from its ensemble structure and advanced regularization. In contrast, the GloVe-based SVM model and the BERT-based Random Forest model achieved lower performance levels, with accuracies of 0.70 and 0.67, respectively. Overall, the comparative analysis underscores not only the flexibility of the ***sleepCare*** framework to accommodate different feature extraction and classification strategies but also the critical importance of addressing class imbalance to enhance the robustness and fairness of sleep disorder classification models.

The superior performance of the Unigram-based Naive Bayes model compared to higher-order n-gram models can be attributed to the nature of sleep narrative texts, which are relatively short and structured, making individual word-level features highly informative [38]. As the n-gram size increases, feature sparsity becomes more pronounced, leading to poorer generalization due to the rarity of longer phrases within a moderate-sized dataset [39]. Similarly, the GloVe-based SVM model performed worse than the Unigram model, likely because GloVe embeddings, while capturing semantic similarities, may dilute fine-grained distinctions between closely related sleep symptoms, which are critical for accurate classification [40]. In contrast, the BERT-based models leveraged deep contextual embeddings that preserved subtle linguistic cues within the narratives, resulting in better performance overall [30]. Among the BERT-based classifiers, the Support Vector Machine model outperformed Random Forest and XGBoost classifiers, possibly due to its ability to handle high-dimensional and sparse representations more effectively [41,42]. SVMs are known to perform particularly well when the number of features greatly exceeds the number of samples, as is typical when using dense BERT embeddings. Additionally, the incorporation of class weighting in the SVM tuning process further enhanced its robustness across minority classes, while ensemble-based methods like Random Forest and XGBoost may have struggled with overfitting or bias toward majority classes despite their flexibility.

While the ***sleepCare*** pipeline demonstrates strong potential for classifying sleep-related narratives, several limitations must be acknowledged. First, the size of the curated dataset, although carefully constructed, remains relatively modest, potentially limiting the generalizability of the models to broader populations and more diverse linguistic styles [43]. Second, the current system relies on supervised learning, requiring labeled data for training, which may not scale easily without additional annotation efforts [44]. Furthermore, while the ***sleepCare*** pipeline addresses class imbalance to some extent through weighting strategies, more advanced techniques such as synthetic data generation or adaptive sampling could be explored to further enhance performance, especially in minority classes [22,45]. Another limitation is that the classification is based solely on textual input without the integration of multimodal sleep-related data (e.g., actigraphy, polysomnography summaries), which could enrich the context and improve diagnostic accuracy [46,47]. Future work will focus on expanding the dataset to include a wider range of sleep disorder narratives, exploring semi-supervised or self-supervised learning approaches to reduce labeling costs [44], and integrating multimodal features to support more comprehensive and explainable models [46]. Additionally, deployment in real-world rural and remote healthcare environments, accompanied by usability and feasibility studies, will be essential to validate the practical impact of the ***sleepCare*** framework [48,49].

The development of the ***sleepCare*** pipeline marks the first step toward establishing a broader roadmap for integrating natural language processing technologies into sleep medicine. By systematically categorizing free-text sleep narratives, the approach creates a structured foundation for building intelligent systems capable of assisting clinical decision-making. Future iterations of ***sleepCare*** will evolve beyond static multi-class classification to support predictive modeling, risk stratification, and symptom progression tracking, enabling earlier and more personalized interventions. A key direction involves the development of a conversational agent or chatbot that provides personalized recommendations, screens for sleep disorders, and delivers continuous 24/7 support, especially valuable in rural or underserved areas. To support this, ***sleepCare*** will adopt a closed-loop feedback system, allowing it to learn dynamically from user inputs, clinician feedback, and misclassification patterns. This feedback-driven architecture will ensure that model updates are informed by real-world performance, improving accuracy, equity, and robustness over time. Future enhancements will also include expanding the dataset with diverse, lived-experience narratives, integrating semi-supervised learning to reduce annotation needs, and fusing multimodal data (e.g., wearables, mobile apps) for richer context. As ***sleepCare*** matures, it will facilitate automatic triaging, identify high-risk individuals, and support referral prioritization. Ultimately, its ability to translate subjective narratives into clinically meaningful outputs positions ***sleepCare*** as a scalable and explainable decision-support tool, complementing traditional diagnostics and advancing the digital transformation of sleep healthcare.

To sum up, the ***sleepCare*** pipeline establishes a scalable and adaptable foundation for applying natural language processing to sleep health. By combining methodological rigor with a focus on accessibility, transparency, and clinical relevance, this work represents a significant step toward more intelligent, equitable, and patient-centered sleep medicine. Future developments will aim to expand the system’s capabilities, enhance real-world deployment, and further integrate personalized support tools into digital health ecosystems.

## 5. Conclusions

The current study introduced the ***sleepCare*** pipeline, an automated natural language processing framework designed to classify sleep-related narratives with high accuracy, transparency, and adaptability. By combining traditional n-gram models (n = 1…9), pre-trained word embeddings such as GloVe and BERT, and a range of machine learning (Support Vector Machines, Random Forests, and XGBoost) classifiers, ***sleepCare*** demonstrates the potential of text-based approaches to support the early identification of sleep disorders. Through rigorous preprocessing, advanced feature extraction, hyperparameter optimization, and model evaluation, the pipeline achieves strong and balanced classification performance across different symptom categories. The open release of a curated, clinically aligned dataset further promotes reproducibility, scientific collaboration, and the development of explainable AI tools in sleep research. Taken together, these contributions provide an important foundation for advancing digital sleep health interventions, particularly in rural and underserved communities where traditional diagnostic resources remain limited.

## Figures and Tables

**Figure 1 brainsci-15-00667-f001:**
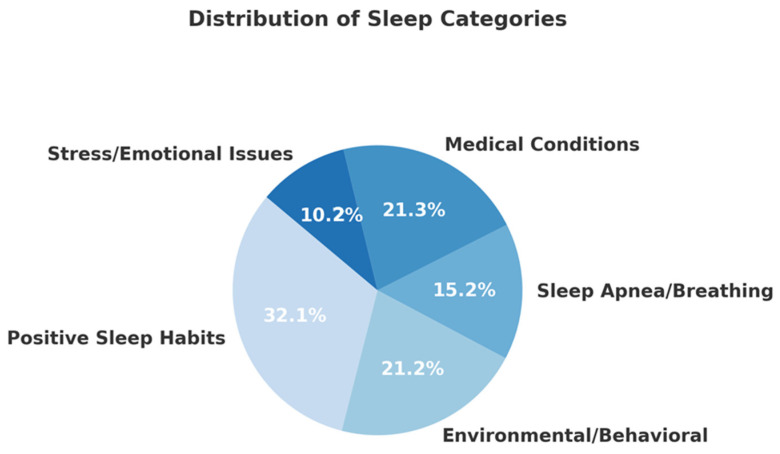
Distribution of the sleep narratives across the five (5) different categories.

**Figure 2 brainsci-15-00667-f002:**
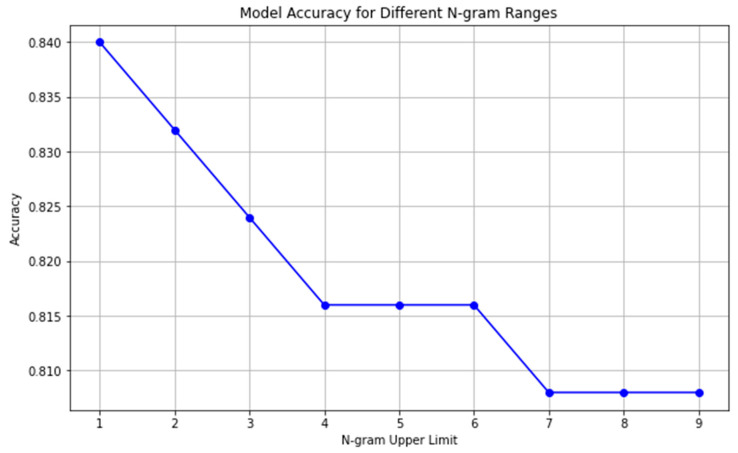
Visualization of accuracy distribution across the different n = 1…9 values.

**Figure 3 brainsci-15-00667-f003:**
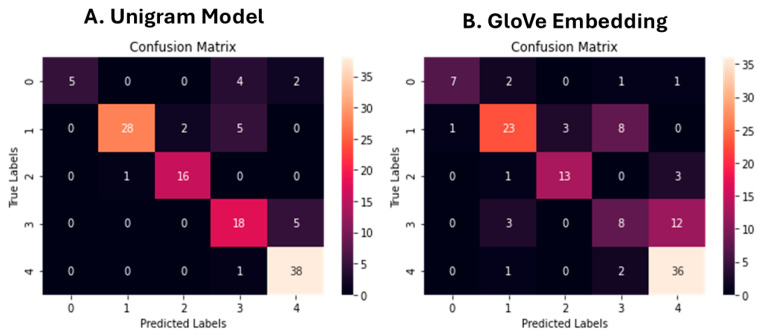
Confusion matrices for the Multinomial Naïve Bayes unigram model (**A**) and for the GloVe word embedding using Support Vector Machines (**B**).

**Figure 4 brainsci-15-00667-f004:**
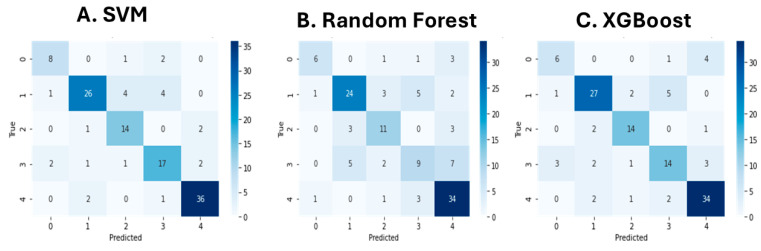
Confusion matrices for the BERT transformers pipeline using Support Vector Machines (**A**), Random Forests (**B**), and XGBoost (**C**) classifiers.

**Table 1 brainsci-15-00667-t001:** Multinomial Naive Bayes accuracy across different (n = 1…9) n-gram configurations.

n-Value	Accuracy
1	0.840
2	0.832
3	0.824
4	0.816
5	0.816
6	0.816
7	0.808
8	0.808
9	0.808

**Table 2 brainsci-15-00667-t002:** Evaluation metrics (precision, recall, f1-score, and support) for the unigram model.

Class	Precision	Recall	F1-Score	Support
0	1.00	0.45	0.62	11
1	0.97	0.80	0.88	35
2	0.89	0.94	0.91	17
3	0.64	0.78	0.71	23
4	0.84	0.97	0.90	39

**Table 4 brainsci-15-00667-t004:** Classification accuracy of the BERT Transformer wrapped with an SVM classifier.

Class	Precision	Recall	F1-Score	Support
0	0.73	0.73	0.73	11
1	0.87	0.74	0.80	35
2	0.70	0.82	0.76	17
3	0.71	0.74	0.72	23
4	0.90	0.92	0.91	39

**Table 5 brainsci-15-00667-t005:** Classification accuracy of the BERT Transformer wrapped with a Random Forest classifier.

Class	Precision	Recall	F1-Score	Support
0	0.75	0.55	0.63	11
1	0.75	0.69	0.72	35
2	0.61	0.65	0.63	17
3	0.50	0.39	0.44	23
4	0.69	0.87	0.77	39

**Table 6 brainsci-15-00667-t006:** Classification accuracy of the BERT Transformer wrapped with an XGBoost classifier.

Class	Precision	Recall	F1-Score	Support
0	0.60	0.55	0.57	11
1	0.82	0.77	0.79	35
2	0.78	0.82	0.80	17
3	0.64	0.61	0.62	23
4	0.81	0.87	0.84	39

**Table 7 brainsci-15-00667-t007:** Summarization of the comparative analysis of the five (5) models deployed in the sleep text narrative analysis.

Model	Accuracy	Macro Precision	Macro Recall	Macro F1-Score
**Unigram + Naive Bayes**	0.84	0.87	0.79	0.80
**GloVe + SVM (Polynomial kernel)**	0.70	0.71	0.67	0.68
**BERT + SVM (RBF kernel)**	0.81	0.78	0.79	0.78
**BERT + Random Forest**	0.67	0.66	0.63	0.64
**BERT + XGBoost**	0.76	0.73	0.72	0.73

## Data Availability

Dataset and code available at [29] https://github.com/cfrantzidis/sleepCare (27 April 2025).

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
