# Peer review of "From Narratives to Diagnosis: A Machine Learning Framework for Classifying Sleep Disorders in Aging Populations: The sleepCare Platform"

_brainsci, 2025, doi:10.3390/brainsci15070667_

Round 1

Reviewer 1 Report

Comments and Suggestions for Authors

I found this thoughtful application of AI and ML very interesting and potentially useful for this use case, and its scalability to others. While I was a bit disappointed that shiftwork, and its imposed irregularity of the sleep/wake cycle, and subsequent impact on the endocrine system, etc. were not more pronounced is the write up, the level of detailed on the compared methodologies was a real bonus. 

Most of my comments are grammatical, which in itself speaks to the rigor by which the paper was composed. These include:

  1. Line 73: "Despite its limited" is a fragment and should be removed
  2. Line 75: Missing a word between First and Using.
  3. Line 90: "...evidence OF the association..."
  4. Line 107: remove "So"
  5. Line 108: remove "thus"
  6. Line 110: "improveD"
  7. Lines 138/139: you talk about irregular schedules but do not define them or talk about how shiftwork influences those - presumably - sleep wakes schedules. I feel a bit more in the intro could be said about this as the shiftwork and sleep literature is mature - 50 years old+.
  8. Line 142: "The dataset consists OF....with STRESS/emotional issues...
  9. Lines 144/145: when talking about the poor sleep habits attributed to environmental factors would be a good place to discuss shiftwork, forcing sleep outside of normal ranges, and reducing its quality and duration.
  10. A little more detail on the size of eacch of the sample bins would be helpful - why were you satisfied with this split. Also the pie chart does not add much value and the colors are very distracting.
  11.  Line 207:  ....we SWITCHED...
  12. Line 226: We next implemented a more...
  13. Line 245 - replace Held-Out with sequestered 
  14. Line 266: "Reaches" sounds odd here; consider rewording. 
  15. Line 271: delete sentence starting "Regarding the..."
  16. LIne 328: It would be helpful to understand if the author feels these changes in accuracy are meaningful, and not just different. 
  17. Line 492: "access to SPECIALIZED sleep..."
  18. Line 493: so how to evaluate whether sleepCare enhances the set of digital health tools? This seems like a very nebulous statement. In what way, how will it be compared to other tools, etc.?
  19. There are other sleep disorders screening tools available that people just need to answer a few questions to get a sense if they have a problem. How does this solution compared to those fatigue models/algorithms relying to clinical data and relates patient/user questioning? There are a number of such solutions with some literature on them. 
  20. How does sleepCare learn from its failures and becomes better?

Author Response

Reviewer 1

  1. Line 73: "Despite its limited" is a fragment and should be removed

I would like to thank the reviewer for that typo. We removed it from the current version of the manuscript.

  1. Line 75: Missing a word between First and Using.

Aiming to improve the readability of this sentence, we rephrased it as follows: Although being a pioneering NLP study in the fields of sleep-related texyal data, it suffered from several limitations especially when dealing with ambiguous terminology

  1. Line 90: "...evidence OF the association..."

Thanks for highlighting this syntactic typo. This was corrected as follows:

Although there is concrete evidence of the association of sleep disorders with cognitive decline, physical and mental wellbeing, the identification of sleep disorders is mainly performed in clinical environments through polysomnography [19] or in real-world settings with wearables and or mobile applications [20].

  1. Line 107: remove "So"

I would like thank the reviewer for this. We deleted the word ‘So’ in the beginning of this paragraph

  1. Line 108: remove "thus"

Done.

  1. Line 110: "improveD"

I would like to thank the reviewer for that comment. Indeed, we employed past tense and the entire paragraph rephrased as follows:

The proposed pipeline represents a scalable, non-invasive, and efficient alternative enabling the early detection and differentiation of sleep problems across diverse and/or remote populations. Firstly, we investigated baseline models founded on naïve techniques and then improved the classification accuracy employing word embeddings and transformer-based models. The latter concepts improved the robustness of our approach, making it capable of processing unstructured textual input derived from patient self-reports, electronic health records, chatbot conversations and classifying them into five clinically meaningful categories. These categories not only differentiated between normal and disordered sleep but also identified probable etiologies: (1) stress-related factors, (2) neurodegenerative processes, (3) breathing-related disorders, and (4) poor sleep habits.

  1. Lines 138/139: you talk about irregular schedules but do not define them or talk about how shiftwork influences those - presumably - sleep wakes schedules. I feel a bit more in the intro could be said about this as the shiftwork and sleep literature is mature - 50 years old+.

This is a very important comment. The reviewer successfully identified a gap in the Introduction section. Aiming to reply to the reviewer’s comment we add the following lines (59-64): There is concrete evidence that irregular work schedules and shiftwork induce sleep disorders in shift-working nurses. A previous study employed the Bergen Shft Work Sleep Questionnaire (BSWSQ) to survey 1,586 Norwegian nurses and found that night shifts and rotating schedules were most associated with insomnia and sleepiness. Moreover, three-shift rotations induced higher insomnia rate than permanent night shifts [14].

  1. Line 142: "The dataset consists OF....with STRESS/emotional issues...

I apologize to the reviewer for those typos. The sentence was corrected as follows: The dataset consists of 1) 43 instances (6.9%) of sleep disorders with stress/emotional

  1. Lines 144/145: when talking about the poor sleep habits attributed to environmental factors would be a good place to discuss shiftwork, forcing sleep outside of normal ranges, and reducing its quality and duration.

Apart from adding an introductory section (see reply associated with the 7th comment), we also added the following in this part: 98 instances (15.7%) of poor sleep habits attributed to environmental/behavioral factors in accorddance with a study investigating shiftwork in Norwegian nurses [14]

  1. A little more detail on the size of eacch of the sample bins would be helpful - why were you satisfied with this split. Also the pie chart does not add much value and the colors are very distracting.

This is a very crucial point that indeed needed improvement. I tried to clarify the issues associated with the dataset generation by adding the following:

The dataset aims to integrate the scientific evidence associated with good sleep practices and sleep-associated disorders as defined by world-leading organizations [24-28], Then, it transforms those guidelines into patient-related narratives. It is an ongoing procedure that is currently not balanced across all the classification categories. Therefore, there is need for further enhancing the sleep disorders and especially those associated with mental health factors. Further dataset versions should also integrate the existing surrogate gate with real-world instances from lived experience. It is publicly available at the following link [29] and can be applied to several research and development areas, including:

I apologize for the low readability and the distraction caused due to the color selection. I produced an updated pie chart which features larger, bold font for improved visibility, and a softer, more neutral blue color palette to enhance readability and reduce visual distraction:

  1.  Line 207:  ....we SWITCHED...

Thanks for identifying this typo. The entire sentence rephrased as follows:             The model switched to inference mode by calling the ‘.eval()’ method.

  1. Line 226: We next implemented a more...

Similarly, the sentence was rephrased as following:            A more advanced model was next implemented

  1. Line 245 - replace Held-Out with sequestered 

The replacement took place according to the reviewer’s comment:            After identifying the optimal parameter set, we retrained the SVM model using the best combination on the full training data and evaluated its performance on the sequestered test set.

  1. Line 266: "Reaches" sounds odd here; consider rewording. 

Indeed, the reviewer is right. The sentence was rephrased as follows: It determines the influence extent of each unique training instance

  1. Line 271: delete sentence starting "Regarding the..."

This sentence was deleted according to the reviewer’s comment.

  1. LIne 328: It would be helpful to understand if the author feels these changes in accuracy are meaningful, and not just different. 

This is a very important comment. The reviewer seems very experienced with the evaluation strategy of machine learning models and correctly identified it. Aiming to satisfy the curiosity of a potential expert reader, the following section was added:

However, it should be highlighted that the accuracy drop due to the increase in the n-value was not a dramatic one. The baseline model showed a steady performance which was a bit higher than 80% and was enhanced for the smallest n-values. This finding is further discussed in the remainder of the article and was associated with the small size of the individual narratives.

  1. Line 492: "access to SPECIALIZED sleep..."

The typo was corrected. Thanks for identifying that.

  1. Line 493: so how to evaluate whether sleepCare enhances the set of digital health tools? This seems like a very nebulous statement. In what way, how will it be compared to other tools, etc.?

Although this issue is discussed in the Future work section which contains both limitations and a potential roadmap, the readers may feel a sense of speculation at that specific point of the Discussion section. Aiming to avoid this and further clarify the proposed roadmap, the following paragraph was added at this point:

Thε efficacy of the sleepCare should be further validated through a pilot phase which would contain real-world data. The study design should integrate elderly participants from rural and urban settings which would randomly assigned either to a group interacting with the sleepCare system or with golden standard, clinical settings such as access to sleep clinics equipped with polysomnographic sensors and sleep experts. Potential outcome measures should be the accuracy of both approaches, the ease of use and the users’ satisfaction and adherence.

  1. There are other sleep disorders screening tools available that people just need to answer a few questions to get a sense if they have a problem. How does this solution compared to those fatigue models/algorithms relying to clinical data and relates patient/user questioning? There are a number of such solutions with some literature on them. 

Indeed the reviewer is right. Aiming to respond to this significant comment, we added the following segment in lines 517-521: Apart from those dependent variables, the classification accuracy of the sleepCare system should be further validated by means of sensorial and question-related data. Therefore, the future study design should also investigate whether the syste’'s approach would be statistically significantly correlated with the outcome of the Pittsburgh Sleep Quality Index (PSQI) [36] or the SmartHypnos mobile application [21].

  1. How does sleepCare learn from its failures and becomes better?

This is a very important question that should be answered within the Discussion section. Aiming to reply to the reviewer’s comment, we restructured the concluding part of the manuscript as follows:

The development of the sleepCare pipeline marks the first step toward establishing a broader roadmap for integrating natural language processing technologies into sleep medicine. By systematically categorizing free-text sleep narratives, the approach creates a structured foundation for building intelligent systems capable of assisting clinical decision-making. Future iterations of sleepCare will evolve beyond static multi-class classification to support predictive modeling, risk stratification, and symptom progression tracking, enabling earlier and more personalized interventions. A key direction involves the development of a conversational agent or chatbot that provides personalized recommendations, screens for sleep disorders, and delivers continuous 24/7 support—especially valuable in rural or underserved areas. To support this, sleepCare will adopt a closed-loop feedback system, allowing it to learn dynamically from user inputs, clinician feedback, and misclassification patterns. This feedback-driven architecture will ensure that model updates are informed by real-world performance, improving accuracy, equity, and robustness over time. Future enhancements will also include expanding the dataset with diverse, lived-experience narratives, integrating semi-supervised learning to reduce annotation needs, and fusing multimodal data (e.g., wearables, mobile apps) for richer context. As sleepCare matures, it will facilitate automatic triaging, identify high-risk individuals, and support referral prioritization. Ultimately, its ability to translate subjective narratives into clinically meaningful outputs positions sleepCare as a scalable and explainable decision-support tool, complementing traditional diagnostics and advancing the digital transformation of sleep healthcare.

Reviewer 2 Report

Comments and Suggestions for Authors

This manuscript has conducted detailed data analysis and discussion on Sleep Disorders. The data sources, analysis methods and analysis results in the manuscript are sufficient, but the following three aspects need to be modified:

1. The data of the manuscript is classified by SVM, random forest and other methods. Please specify the key optimization parameters of relevant algorithms, such as the penalty parameters in SVM.
2. The dataset is a key strength, but more detail on how labeling was done (manual vs. automated, inter-rater agreement, etc.) would increase transparency.
3. The error in the confusion matrix obtained by the classification algorithm needs further analysis and discussion.

Author Response

This manuscript has conducted detailed data analysis and discussion on Sleep Disorders. The data sources, analysis methods and analysis results in the manuscript are sufficient, but the following three aspects need to be modified:

  1. The data of the manuscript is classified by SVM, random forest and other methods. Please specify the key optimization parameters of relevant algorithms, such as the penalty parameters in SVM.

We would like to thank the reviewer for the kind words and the constructive feedback. The manuscript has already been revised according to the comments of three other reviewers. Indeed, some of them have already highlighted this issue. So, in the revised manuscript, we made the following changes:

In Section 2.6.1 we highlighted with black color the optimal value for the SVM parameters (lines 269 – 274).

  • C: Regularization parameter — values of [0.1, 0.4, 1, 2, 3, 4]
  • gamma: Kernel coefficient — values of [0.001, 0.01, 0.1, 0.4, 1, 2, 3, 4, ’scale’, ’auto’]
  • kernel: Kernel type — values of ['linear', 'rbf', ‘poly’, ‘sigmoid’]
  • degree of the polynomial kernel – values of [2, 3, 4, 5, 6, 7. 8]
  • ‘class_weight’ : [None, ‘balanced’]

In line 288 we added the following sentence: The optimal parameters are indicated in bold.

In Section 2.6.2 we added the following in lines 307 – 308: The optimal parameter set is the following: {'bootstrap': False, 'max_depth': None, 'min_samples_leaf': 2, 'min_samples_split': 2, 'n_estimators': 30}.

We also restructured 2.6.3 as follows (lines 316-335): Finally, we employed an XGBoost classifier trained on BERT-generated sentence embeddings. The data was partitioned into training and test sets using an 80/20 split, and transformed into XGBoost’s optimized DMatrix format. We applied a structured three-stage grid search approach to identify the best hyperparameter configuration. In Stage 1, tree structure parameters (max_depth and min_child_weight) were tuned using 5-fold cross-validation with early stopping based on multi-class log loss. The optimal values were: 1) max_depth: 3, 2) min_child_weight: 5. In Stage 2, we further refined the model by adjusting regularization (reg_alpha, reg_lambda) and sampling parameters (subsample, colsample_bytree). The optimal values obtained in this stage were: 1) subsample: 0.7, 2) colsample_bytree: 0.7, 3) reg_alpha: 0.5, 4) reg_lambda: 1.5. Finally, in Stage 3, we explored various learning rates (learning_rate) to balance convergence speed and generalization. At each stage, the configuration yielding the lowest average log loss on validation folds was selected. The final model was trained on the full training set using the best parameters and optimal number of boosting rounds. Performance was evaluated on the test set using a classification report, confusion matrix, and learning curve visualization, providing a comprehensive view of the model’s accuracy and calibration across classes. The final set of the fine-tuned parameters is the following: { 'objective': 'multi:softprob', 'num_class': 5, 'eval_metric': 'mlogloss', 'seed': 42, 'max_depth': 3, 'min_child_weight': 5, 'subsample': 0.7, 'colsample_bytree': 0.7, 'reg_alpha': 0.5, 'reg_lambda': 1.5, 'learning_rate': 0.05}}.

  1. The dataset is a key strength, but more detail on how labeling was done (manual vs. automated, inter-rater agreement, etc.) would increase transparency.

Again, please accept my apologies for the inconvenience. This section was restructured as following (lines 160-166):

The dataset aims to integrate scientific evidence associated with good sleep practices and sleep-associated disorders as defined by world-leading organizations [24-28], Then, it transforms those guidelines into patient-related narratives. It is an ongoing procedure that is currently not balanced across all the classification categories. Therefore, there is a need for further enhancing the sleep disorders and especially those associated with mental health factors. Further dataset versions should also integrate the existing surrogate gate with real-world instances from lived experience.

  1. The error in the confusion matrix obtained by the classification algorithm needs further analysis and discussion.

Thanks for indicating this significant issue. The following revisions were performed:

In Section 3.1 (Baseline Model Evaluation) the following text was added in lines 349 354:

However, it should be highlighted that the accuracy drop due to the increase in the n-value was not a dramatic one. The baseline model showed a steady performance which was a bit higher than 80% and was enhanced for the smallest n-values. This finding is further discussed in the remainder of the article and was associated with the small size of the individual narratives.

We also added the accuracy score in line 385: The overall accuracy was 0.70.

Within the Discussion Section we added a justification (lines 507 – 512) regarding the selection of the machine learning models:

The sleepCare pipeline currently employes traditional machine learning algorithms such as the 1) Support Vector Machines, 2) Random Forests and 3) XGBoost. The selection of the machine learning models was performed based on the following criteria: 1) To identify the ones most frequently used in NLP pipelines and 2) To employ models that allow easier hyper-parameter tuning to optimize the model’s performance. As the dataset size increases, more elaborate neural (recurrent) network architectures will be investigated.

We also added in lines 523 – 533 a critical reflection on the future work needed for the validation of the sleepCare approach:

The efficacy of the sleepCare should be further validated through a pilot phase which would contain real-world data. The study design should integrate elderly participants from rural and urban settings which would be randomly assigned either to a group interacting with the sleepCare system or with golden standard, clinical settings such as access to sleep clinics equipped with polysomnographic sensors and sleep experts. Potential outcome measures should be the accuracy of both approaches, the ease of use and the users’ satisfaction and adherence. Apart from those dependent variables, the classification accuracy of the sleepCare system should be further validated by means of sensorial and question-related data. Therefore, the future study design should also investigate whether the syste’'s approach would be statistically significantly correlated with the outcome of the Pittsburgh Sleep Quality Index (PSQI) [36] or the SmartHypnos mobile application [21].

We also updated the roadmap of the future steps that need to be performed (lines 623 – 634):

To support this, sleepCare will adopt a closed-loop feedback system, allowing it to learn dynamically from user inputs, clinician feedback, and misclassification patterns. This feedback-driven architecture will ensure that model updates are informed by real-world performance, improving accuracy, equity, and robustness over time. Future enhancements will also include expanding the dataset with diverse, lived-experience narratives, integrating semi-supervised learning to reduce annotation needs, and fusing multimodal data (e.g., wearables, mobile apps) for richer context. As sleepCare matures, it will facilitate automatic triaging, identify high-risk individuals, and support referral prioritization. Ultimately, its ability to translate subjective narratives into clinically meaningful outputs positions sleepCare as a scalable and explainable decision-support tool, complementing traditional diagnostics and advancing the digital transformation of sleep healthcare.

The author remains at the disposal of the reviewer for any further comments that need to be addressed.

Reviewer 3 Report

Comments and Suggestions for Authors

The research presented in the manuscript is valuable and contributes to the field. However, the writing quality is not good and does not meet the academic publication standard. Many sentences in the introduction and some other sections are not clear and grammatically incorrect. It is very hard to understand the content. There are many sentences with unclear pronoun references and phrasing that may confuse readers.

1. Authors should explain why particular machine learning models (SVM, RF, and XGBoost) were chosen over others.
2. Authors should clarify how variability in patients’ language and vocabulary (especially among aging populations) was standardized before feature extraction.
3. There is no detail on how the ground-truth labels for sleep disorder were verified and annotated. Were any sleep specialists involved in this process?
4. There is no information on how features were selected for model input, and also about dimensionality reduction. Did the authors use any dimensionality reduction method, like PCA, LASSO, to mitigate overfitting?
5. Can the authors elaborate on the handling of negation in clinical narratives, “no history of insomnia"? Does any negation detection algorithm or rule-based logic integrated into the preprocessing pipeline?
6. Did the authors perform any feature importance or SHAP analysis to interpret which narrative features contributed most to classification decisions?
7. Clarification on how hyperparameters were tuned is required. Mention which techniques were used, and also report the optimal values used.
8. Are there any existing systems or literature that use machine learning models and narrative data for classifying sleep disorders in aging adults? A structured comparison should be detailed in the paper discussing the proposed method and related work in the literature.

Author Response

The research presented in the manuscript is valuable and contributes to the field. However, the writing quality is not good and does not meet the academic publication standard. Many sentences in the introduction and some other sections are not clear and grammatically incorrect. It is very hard to understand the content. There are many sentences with unclear pronoun references and phrasing that may confuse readers.

Aiming to address the reviewer’s comment, the manuscript was carefully revised and all grammatical and syntactical errors were identified and corrected.

  1. Authors should explain why particular machine learning models (SVM, RF, and XGBoost) were chosen over others.

The selection of the machine learning models was performed based on the following criteria: 1) To identify the ones most frequently used in NLP pipelines and 2) Employ models that allow easier hyper-parameter tuning to optimize the model’s performance. As the dataset size will increase, we would investigate neural (recurrent) network architectures.

  1. Authors should clarify how variability in patients’ language and vocabulary (especially among aging populations) was standardized before feature extraction.

In section 2.1 we clearly explain that the dataset consists of surrogate data and do not involve any patients at all. However, the reviewer addresses a very serious comment that should be addressed when dealing with real-world data. Any NLP pipeline should fuse different covabulary settings (formal, informal ones) aiming to eliminate any bias or discrimination effect.

  1. There is no detail on how the ground-truth labels for sleep disorder were verified and annotated. Were any sleep specialists involved in this process?

No sleep specialists were directly involved in this procedure. We extracted the information material about good sleep and the main sleep disorders from the most common, valid and prestigious electronic sites (e.g. National Institute of Health, American Academy of Sleep Medicine) and we transformed them as sleep narratives of one sentence.

The author is a sleep expert with over a decade experience in sleep acquisition and analysis.

The reviewer may find the following text useful:

We constructed a dataset consisting of narrative, sleep-related text samples in sentence form. The text samples were derived from electronic sources describing the causes and factors of the main sleep disorders, such as the National Sleep Foundation (NSF) [24], the American Academy of Sleep Medicine (AASM) [25], Centers for Disease Control and Prevention (CDC) – Sleep and Sleep Disorders [26], National Institutes of Health (NIH) – Sleep Disorders Information [27] and World Sleep Society [28]. Each sentence is labeled according to the underlying reason for sleep quality or disturbance. The dataset contains a total of 623 text entries, where each entry is a free-form narrative describing an individual's sleep experience.

  1. There is no information on how features were selected for model input, and also about dimensionality reduction. Did the authors use any dimensionality reduction method, like PCA, LASSO, to mitigate overfitting?

Word embedding models transform the text into vector embeddings. These are the features used. The pipeline used the word embeddings and did not perform any engineering procedure on them. Aiming to avoid overfitting we split the dataset into training and test set. We also investigated hyper-parameter tuning.

  1. Can the authors elaborate on the handling of negation in clinical narratives, “no history of insomnia"? Does any negation detection algorithm or rule-based logic integrated into the preprocessing pipeline?

We should bear in mind that we currently used as input surrogate textual data. So, the “healthy” sleep category was the one reflecting a good sleep habit accompanied by the absence of any sleep disorder.

  1. Did the authors perform any feature importance or SHAP analysis to interpret which narrative features contributed most to classification decisions?

We used both static (word embeddings) and the BERT transformer without performing any feature selection on them. A SHAP analysis may be useful in a latter case to provide a glass-box architecture.

  1. Clarification on how hyperparameters were tuned is required. Mention which techniques were used, and also report the optimal values used.

As indicated within Section 2.6, the GridSearchCV method is used.

Thanks for indicating this missing information. Regarding the SVM we highlighted the optimal values in bold:

  • C: Regularization parameter — values of [0.1, 0.4, 1, 2, 3, 4]
  • gamma: Kernel coefficient — values of [0.001, 0.01, 0.1, 0.4, 1, 2, 3, 4, ’scale’, ’auto’]
  • kernel: Kernel type — values of ['linear', 'rbf', ‘poly’, ‘sigmoid’]
  • degree of the polynomial kernel – values of [2, 3, 4, 5, 6, 7. 8]
  • ‘class_weight’ : [None, ‘balanced’]

We also included the following sentence:  The optimal parameters are indicated in bold.

Regarding the Random Forest Classifier, we added the following: The optimal parameter set is the following: {'bootstrap': False, 'max_depth': None, 'min_samples_leaf': 2, 'min_samples_split': 2, 'n_estimators': 30}.

Regarding the XGBoost classifier, we revised that section as follows:

Finally, we employed an XGBoost classifier trained on BERT-generated sentence embeddings. The data was partitioned into training and test sets using an 80/20 split, and transformed into XGBoost’s optimized DMatrix format. We applied a structured three-stage grid search approach to identify the best hyperparameter configuration. In Stage 1, tree structure parameters (max_depth and min_child_weight) were tuned using 5-fold cross-validation with early stopping based on multi-class log loss. The optimal values were: 1) max_depth: 3, 2) min_child_weight: 5. In Stage 2, we further refined the model by adjusting regularization (reg_alpha, reg_lambda) and sampling parameters (subsample, colsample_bytree). The optimal values obtained in this stage were: 1) subsample: 0.7, 2) colsample_bytree: 0.7, 3) reg_alpha: 0.5, 4) reg_lambda: 1.5. Finally, in Stage 3, we explored various learning rates (learning_rate) to balance convergence speed and generalization. At each stage, the configuration yielding the lowest average log loss on validation folds was selected. The final model was trained on the full training set using the best parameters and optimal number of boosting rounds. Performance was evaluated on the test set using a classification report, confusion matrix, and learning curve visualization, providing a comprehensive view of the model’s accuracy and calibration across classes. The final set of the fine-tuned parameters is the following: { 'objective': 'multi:softprob', 'num_class': 5, 'eval_metric': 'mlogloss', 'seed': 42, 'max_depth': 3, 'min_child_weight': 5, 'subsample': 0.7, 'colsample_bytree': 0.7, 'reg_alpha': 0.5, 'reg_lambda': 1.5, 'learning_rate': 0.05}}.

  1. Are there any existing systems or literature that use machine learning models and narrative data for classifying sleep disorders in aging adults? A structured comparison should be detailed in the paper discussing the proposed method and related work in the literature.

This is a very important question. Currently, to the best of my knowledge, there are no system that is based on narrative data. Our group has previously published articles performing sleep stage classification [22] and we also developed the SmartHypnos mobile application [21].

Reviewer 4 Report

Comments and Suggestions for Authors

The author needs to elaborate on what is the significance of sleep disorders in aging populations in the introduction section .

The author needs to explain how is your proposed system is better than the existing system in section 3 (3. Results)

There is no where the authors mentioned that SVM, RF, and XGBoost are chosen? What is the motivation behind it? There are a lot of good classification algorithms are there in ML. Authors must justify in section 3 

Little bit of curiosity about feature selection in this article. How are the features selected? 

In subsection 2.1. Dataset, the author mentioned that they have constructed a dataset consisting of narrative, sleep-related text samples in sentence form. But my question is, where is the source of the dataset? who has authenticated? What was the sample size taken during the experiment? 

In tables 2,3 etc, the author considered the performance metrics (precision, recall and f1-score) why? why is accuracy not considered? Proper justification required

Like this, many areas are there where the author can improve 

Author Response

  1. The author needs to elaborate on what is the significance of sleep disorders in aging populations in the introduction section .

The significance of sleep disorders in aging is described in the following section:

As we age there are sleep architecture alterations such as reduced slow wave (N3) and Rapid Eye Movement (REM) duration which is often accompanied by more frequent awakenings and increased sleep latency [10]. Those disturbances are commonly associated with comorbidities such as chronic conditions (cardiovascular disease, diabetes, chronic pain) [6], psychiatric disorders (depression, anxiety) [11] and lifestyle factors such as sedentary behavior and lack of activity [12]. Importantly, emerging evidence suggests that sleep disturbances may precede and potentially contribute to the development of neurodegenerative disorders by accelerating the accumulation of pathological proteins (e.g., amyloid-beta and tau in AD) and impairing the clearance mechanisms such as glymphatic flow [13].

  1. The author needs to explain how is your proposed system is better than the existing system in section 3 (3. Results)

There may be a confusion here. There is no (to the best of the author’s knowledge) existing system that performs sleep text classification. The pipeline refered as a baseline model, is the first model that was developed in this study. It is based on a Naïve Bayes classifier employing n-values form n=1…9. There is a detailed discussion, why the BERT Transformer with Support Vector Machines is better than the baseline model.

  1. There is no where the authors mentioned that SVM, RF, and XGBoost are chosen? What is the motivation behind it? There are a lot of good classification algorithms are there in ML. Authors must justify in section 3 

The sleepCare pipeline currently employes traditional machine learning algorithms such as the 1) Support Vector Machines, 2) Random Forests and 3) XGBoost. The selection of the machine learning models was performed based on the following criteria: 1) To identify the ones most frequently used in NLP pipelines and 2) To employ models that allow easier hyper-parameter tuning to optimize the model’s performance. As the dataset size will increase, more elaborate neural (recurrent) network architectures would be investigated.

  1. Little bit of curiosity about feature selection in this article. How are the features selected? 

Word embedding models transform the text into vector embeddings. These are the features used. The pipeline used the word embeddings and did not perform any engineering procedure on them. Aiming to avoid overfitting we split the dataset into training and test set. We also investigated hyper-parameter tuning.

We also used dynamic word embeddings (BERT transformer) without performing any feature selection on them. Feature selection on those pre-trained models goes beyond the scope of this manuscript and may result in degraded procedure. However, it is a very innovative and useful idea for a future funding proposa.

  1. In subsection 2.1. Dataset, the author mentioned that they have constructed a dataset consisting of narrative, sleep-related text samples in sentence form. But my question is, where is the source of the dataset? who has authenticated? What was the sample size taken during the experiment? 

If the reviewer means as “source”, the data repository, it would be accessible upon the manuscript publication on [29]. It would be a publicly available dataset.

Regarding the authentication of the dataset, the reviewer may find the following information useful:

The text samples were derived from electronic sources describing the causes and factors of the main sleep disorders, such as the National Sleep Foundation (NSF) [24], the American Academy of Sleep Medicine (AASM) [25], Centers for Disease Control and Prevention (CDC) – Sleep and Sleep Disorders [26], National Institutes of Health (NIH) – Sleep Disorders Information [27] and World Sleep Society [28].

The dataset aims to integrate the scientific evidence associated with good sleep practices and sleep-associated disorders as defined by world-leading organizations [24-28], Then, it transforms those guidelines into patient-related narratives. It is an ongoing procedure that is currently not balanced across all the classification categories. Therefore, there is need for further enhancing the sleep disorders and especially those associated with mental health factors. Further dataset versions should also integrate the existing surrogate gate with real-world instances from lived experience.

  1. In tables 2,3 etc, the author considered the performance metrics (precision, recall and f1-score) why? why is accuracy not considered? Proper justification required

Table 2 contains the evaluation metrics (precision, recall, f1-score) for the various (n=1…9) n-gram models. The accuracy value for each model is reported in Table 1.

Regarding the GloVe word embedding pipeline, associated with Table 3, we added the following line : The overall accuracy was 0.70.

In general accuracy does not work well with imbalance datasets. However, it could be easily estimated through the information reported in the confusion matrices.

  1. Like this, many areas are there where the author can improve 

Thanks for the constructive feedback. The manuscript was carefully revised. If the reviewer wishes to suggest any other specific modifications, the author would revise the manuscript accordingly.

Round 2

Reviewer 2 Report

Comments and Suggestions for Authors

The authors have responded to my concerns

Author Response

We want to thank the reviewer for the kind words and the comments that helped to improve the quality of the submitted manuscript.

Reviewer 3 Report

Comments and Suggestions for Authors

The authors addressed the comments, and the manuscript presentation has improved. 

Author Response

(The authors gave the same response as above.)

Reviewer 4 Report

Comments and Suggestions for Authors

The author has effectively addressed all the raised comments; however, some improvements are required 

Still, i am not convinced by the statements about the dataset. Before publishing the paper, the author needs to present the link to the dataset 

In Table 7. Summarization of the comparative analysis of the five (5) models deployed in the sleep text narrative analysis, the author suggested to write the algorithms or pseudocode for the hybrid algorithms of the followings

1.GloVe + SVM (Polynomial kernel)

2. BERT + SVM (RBF kernel) 

3.BERT + Random Forest

4. BERT + XGBoos

The author needs to compute the total execution time for the above-mentioned hybrid models . So that the author will reach a solid conclusion 

Author Response

Q1. The author has effectively addressed all the raised comments; however, some improvements are required

A1. We would like to thank the reviewer for the kind words and the comments that helped to improve the quality of the submitted manuscript.

Q2. Still, i am not convinced by the statements about the dataset. Before publishing the paper, the author needs to present the link to the dataset

A2. Maybe the reviewer has missed that information but there is the following GitHub repository that contains the entire code and the dataset used in the manuscript. The link is cited within the document as Reference 29 and is the following one: https://github.com/cfrantzidis/sleepCare

Everyone has access to the entire code used and the various dataset versions.

Q3. In Table 7. Summarization of the comparative analysis of the five (5) models deployed in the sleep text narrative analysis, the author suggested to write the algorithms or pseudocode for the hybrid algorithms of the followings 1.GloVe + SVM (Polynomial kernel) 2. BERT + SVM (RBF kernel) 3.BERT + Random Forest 4. BERT + XGBoos

A3. As mentioned before there is free and unlimited access to the code and the dataset provided through the GitHub repository. We guarantee that this will open so as to foster the future research efforts in sleep text classification.

Q4. The author needs to compute the total execution time for the above-mentioned hybrid models . So that the author will reach a solid conclusion

A4. Thanks for pointing this out. We ncluded the following in lines 495-499:

The code was executed on a Dell computer equipped with an Intel i7 processor and a GPU. The execution time for the simpler model, the Naive Bayes classifier, was 0.853 seconds, while the BERT-based methodologies required approximately 15 minutes.